# Non-Coding, RNAPII-Dependent Transcription at the Promoters of rRNA Genes Regulates Their Chromatin State in *S. cerevisiae*

**DOI:** 10.3390/ncrna7030041

**Published:** 2021-07-11

**Authors:** Emma Lesage, Jorge Perez-Fernandez, Sophie Queille, Christophe Dez, Olivier Gadal, Marta Kwapisz

**Affiliations:** Laboratoire de Biologie Moléculaire, Cellulaire et du Développement (MCD), Centre de Biologie Intégrative (CBI), Université de Toulouse, CNRS, UPS, 31062 Toulouse, France; emma.lesage-chevillon@univ-tlse3.fr (E.L.); jorge.perezfernandez@ujaen.es (J.P.-F.); sophie.queille@univ-tlse3.fr (S.Q.); christophe.dez@univ-tlse3.fr (C.D.)

**Keywords:** UPS, RNA polymerase I (RNAPI), RNA polymerase II (RNAPII), ribosomal DNA (rDNA), ribosomal RNA (rRNA), long non-coding RNA (lncRNA)

## Abstract

Pervasive transcription is widespread in eukaryotes, generating large families of non-coding RNAs. Such pervasive transcription is a key player in the regulatory pathways controlling chromatin state and gene expression. Here, we describe long non-coding RNAs generated from the ribosomal RNA gene promoter called UPStream-initiating transcripts (UPS). In yeast, rDNA genes are organized in tandem repeats in at least two different chromatin states, either transcribed and largely depleted of nucleosomes (open) or assembled in regular arrays of nucleosomes (closed). The production of UPS transcripts by RNA Polymerase II from endogenous rDNA genes was initially documented in mutants defective for rRNA production by RNA polymerase I. We show here that UPS are produced in wild-type cells from closed rDNA genes but are hidden within the enormous production of rRNA. UPS levels are increased when rDNA chromatin states are modified at high temperatures or entering/leaving quiescence. We discuss their role in the regulation of rDNA chromatin states and rRNA production.

## 1. Introduction

Non-coding RNA (ncRNA) are key regulators in numerous chromatin-based processes. Both silent chromatin (heterochromatin) and active gene transcription can be regulated by specific ncRNAs [1]. Extremely active gene transcription and heterochromatin domains are both present in ribosomal RNA genes (ribosomal DNA—rDNA locus). The yeast genome *Saccharomyces cerevisiae* contains a single rDNA cluster organized as a linear array of 100–150 repeats on chromosome XII (Figure 1A). Each rDNA repeat consists of the 35S and 5S ribosomal RNA (rRNA) transcribed by RNA polymerase I (RNAPI) and RNA polymerase III (RNAPIII), respectively. The 35S and 5S genes are separated by intergenic spacer regions IGS1 and IGS2, organized in heterochromatin domains [2]. Cryptic ncRNAs originated from IGS1 have been involved in heterochromatin formation, genome stability, and aging [3,4,5]. Despite rRNA synthesis being limiting for cell growth, at any given time, only a fraction of rDNA copies is transcribed by RNAPI (open), the rest are transcriptionally silent and assembled into nucleosomes (closed) [6,7,8]. Open rRNA genes are heavily occupied by transcribing RNAPI [9]. The ratio of open to closed repeats varies depending on the cell cycle stage, on the cell type, and its physiological status, but data from mammalian and yeast cells show that the silencing of rRNA genes does not strictly correlate with the decrease of the amount of rRNA and is not simply a consequence of the absence of RNAPI transcription. Thus, the rDNA chromatin state is regulated through complex and still not well-understood mechanisms.

In mammals, long non-coding transcripts (lncRNAs) have been proposed to regulate ribosomal RNA production and nucleolar integrity [10,11,12,13]. Challenging the cellular equilibrium with high temperatures, starvation, or osmotic stress revealed novel lncRNAs—PAPAS, produced by RNAPII in antisense to rRNA genes. PAPAS targets RNAPI machinery and modifies the epigenetic signature of the rDNA promoter through the recruitment of the chromatin remodeling complex [14], which consequently represses rRNA genes. In serum-deprived cells, PAPAS up-regulation is accompanied by the recruitment of the histone methyltransferase Suv4-20h2, which deposits H4K20me3 at rDNA, repressing its transcription [15]. Additionally, lncRNAs produced by RNAPII generate R-loops, preventing RNAPI from producing sense intergenic non-coding RNAs that can disrupt nucleolar organization [11].

In yeast, cryptic RNAPII promoters are present at the rDNA RNAPI-specific promoter [16] as demonstrated in respiratory-deficient cells [17,18], in mutants affecting the accessory subunit Uaf30 of the Upstream Activating Factor (UAF) [19], or in polymerase switch mutants (PSW) [20]. RNAPII driven rRNA expression was instrumental to explore RNAPI activity. rRNA genes artificially driven by a strong RNAPII promoter allowed mutant lacking RNAPI activity to survive [21]. The Nomura lab made use of such artificial constructs to identify most genes required for rRNA synthesis by RNAPI [22]. RNAPII-dependent transcriptional events were thus studied and characterized in artificial or mutated backgrounds. In light of the novel regulatory roles of ncRNA, we were interested in characterizing these transcripts and addressing their role in wild-type cells.

In this study, we characterized long non-coding RNAs generated from ribosomal the RNA gene promoter, called UPStream-initiating transcripts (UPS). We were able to show that they were produced in wild-type cells at very low levels in exponentially growing cells and were accumulated upon the inhibition of Rat1 exonuclease. They accumulated when rDNA genes were closing, suggesting a possible role in regulating rDNA chromatin structure.

## 2. Results

### 2.1. Upstream-Initiating Transcripts (UPS) Are Present at Low Level in Wild-Type Yeast Cells

Transcription initiating immediately upstream from the RNAPI transcription start site (TSS, see Figure 1B) was initially detected in PSW (polymerase switch) mutants [20]. These cells lack the UAF complex, an essential RNAPI co-factor in vivo, involved in silencing the cryptic RNAPII promoter in the rDNA. Mutants in the UAF complex have impaired rRNA production by RNAPI [23], which is then instead achieved by RNAPII, resulting in extremely slow growing cells. In the absence of UAF, the synthesis of 35S rRNA by RNAPII comes with the expansion of chromosomal rDNA repeats, improving cell growth [24]. In the absence of Uaf30, a non-essential UAF component, cellular growth is severely delayed and the expression of rRNA is carried out by both RNAPI and RNAPII [19,25,26]. In this mutant, RNAPII initiates rRNA synthesis from a still uncharacterized promoter located 9 to 95 nucleotides upstream from the canonical RNAPI TSS [20]. In PSW strains, rRNAs bearing an RNAPI dependent +1TSS are depleted, and 5′-extended pre-rRNA are prevalent and can form ribosomes. In contrast, the *uaf30*∆ strain contains a small fraction of pre-rRNAs synthesized from upstream initiation while the majority (90%) start at canonical RNAPI TSS [19]. In both PSW and *uaf30*∆ strains, a large fraction of rRNA genes stays in the closed/inactive state, although in *uaf30*∆, a small fraction remains open and is heavily occupied by RNAPI [25,27]. Thus, cell growth might be sustained by the very efficient production of the 35S pre-rRNA from a minority of open copies [28], while the promoters of the closed majority of rRNA genes might serve as a template for RNAPII-dependent upstream initiation [25]. However, we cannot exclude that 5′-extended pre-rRNA transcripts might also result from modified RNAPI transcription.

We wondered about the presence of 5′-extended pre-rRNAs in wild-type cells (WT). In order to visualize these transcripts, we used a primer extension method by the reverse transcription (RT) of RNAs with a specific radiolabeled primer (Figure 1B). Products of the reverse transcription were resolved on a polyacrylamide gel. *uaf30*∆ and PSW strains were used as a positive control for the presence of the 5′-extended rRNAs (Appendix A). We could identify at least 10 different rRNA species with different 5′-ends in PSW strains [20]. As expected, 5′-extended 35S was also detected in *uaf30*∆*,* mostly at positions −36 and −29 but not in WT [20,24]. RT of the 5′-end of 35S rRNA was next quantified using highly sensitive quantitative PCR (qPCR). Due to the various 5′-ends of the extended 35S rRNA produced by RNAPII, we quantified 5′-extended rRNA using an amplicon as close as possible to TSS (Figure 1B). We used the amplicon at position −30 (primers D-C) relative to the canonical +1TSS of RNAPI (primers B-A). In the *uaf30*∆ and PSW strains, 5′-extended 35S pre-rRNAs represent 10% and 45% of 35S pre-RNA, respectively (Figure 1C). Interestingly, in WT cells 5′-extended transcripts are detected at about 10^−4^ over +1 TSS pre-rRNAs, and at about 0.5 × 10^−4^ over *scR1*, the abundant RNA component of the signal recognition particle transcribed by RNA polymerase III (Appendix A). The detection of these 5′-extended 35S rRNAs was highly reproducible and far from background detection (*i.e.*, reaction without reverse transcriptase, see Appendix A). We have called these low abundant transcripts UPS for UPStream-initiating transcripts. The 5′-ends of UPS in WT is unknown since they are detected at −50 and −75 bp upstream from +1 TSS of 35S pre-rRNA. The accumulation of UPS remained unchanged in diploid cells and did not depend on the mating type or the sugar used for culturing (Figure 1D).

We conclude that UPS are non-coding RNAs (ncRNA) of unknown function that are present in low amounts in the wild-type cells of budding yeast.

### 2.2. Characterization of UPS Decay Pathway

Under the optimal growth conditions, the low level of UPS in wild-type cells, is a common feature of various classes of ncRNAs in yeast (SUTs, XUTs, CUTs) [29]. Most of these unstable ncRNAs accumulate in mutants lacking RNA degradation factors (e.g., exosome, TRAMP complex, Xrn1) and in the specific physiological conditions that are relevant for their function, allowing for their detection.

In order to sort UPS in an already characterized group of ncRNAs, we quantified their accumulation in a strain lacking Rrp6, a nuclear subunit of the exosomes that degrades RNAs in 3′-5′ direction, that significantly accumulates Cryptic Unstable Transcripts (CUTs; [30]). We also investigated the deletion mutants *TRF4* and *TRF5*, which encode the subunits of the TRAMP complex, a co-factor of exosomes. The UPS level did not increase significantly in strains that had been mutated for the exosomes and TRAMP complex subunits nor in double mutants *rrp6*∆ *trf4*∆ (Figure 2A). Furthermore, we could not detect a significant variation in UPS accumulation in double mutants *uaf30*∆ *trf4*∆ (Figure 2B) although *uaf30*∆ has a level of UPS that is approximately 200 times that of WT. To exclude the notion that UPS accumulation in various mutants is due to a rDNA copy number variation, we evaluated the rDNA copy number as well as the accumulation level of +1 TSS rRNA in all of the tested mutants (Appendix A).

We conclude that the TRAMP complex and nuclear exosomes are not significant players in the UPS decay pathway. We next tested the *XRN1* deletion mutant, a major 5′-3′ exoribonuclease, which leads to the accumulation of XUTs [31]. We could detect a two-fold increase in the level of UPS in *xrn1*∆ strain (Figure 2A). Xrn1 is mainly present in the cytoplasm and is the counterpart of Rat1, an essential nuclear 5′-3′ exoribonuclease [31]. Next, we studied the involvement of Rat1 on UPS accumulation. We observed that the UPS level drastically increased in the *rat1-1* mutated strain (Figure 2C). To confirm the impact of Rat1 on UPS accumulation, we used a strain expressing *RAT1* gene from the *pMET3* promoter that represses gene expression upon the addition of methionine [32]. Using primer extension (as previously described), we could detect UPS RNA accumulation in strains bearing the *pMET3-RAT1* construct but not in WT cells or the *rrp6*∆ mutant (Figure 2D). According to the non-accumulation of UPS in the *rrp6*∆ mutant, the double mutant *pMET3-RAT1 rrp6*∆ resulted in similar levels of UPS than *pMET-RAT1* alone. Interestingly, compared to the numerous 5′-extended rRNA species detected in PSW strains, only −11 and −42 are significantly accumulated upon Rat1 depletion.

We conclude that UPS accumulation is not affected by the exosomes or TRAMP, but it is moderately affected by Xrn1. Interestingly, the Rat1-dependent 5′-3′ nuclear degradation pathway is the main pathway that regulates UPS accumulation.

### 2.3. UPS Are Produced by RNAPII

RNAPII synthesizes 5′-extended pre-rRNA in mutant strains that are defective in RNAPI activity and the PSW and *uaf30*∆ mutant background [19,20]. To confirm that UPS transcription depends on the RNAPII in cells bearing functional RNAPI transcription machinery, we used two different RNAPII mutants: *rpb1-1* and *rpb4*∆ [33,34]. In the *rpb1-1* strain, RNAPII-dependent transcription is less efficient when compared to the wild type and additionally inhibited at 37 °C [33]. As shown in Figure 3A, the level of transcripts initiating upstream from +1 TSS decreased 2-fold in the *rpb1-1* mutant after 60 min at 37 °C compared to when the temperature was set at 25 °C. This result indicated that RNAPII synthesizes the major fraction of UPS. Furthermore, we also observed a strong reduction in the UPS level in the *rpb4*∆ strain (Figure 3A). We also measured the level of UPS in an RNAPI mutant, the *rpa49*∆ strain, a strain defective in rRNA synthesis and with a slow-growth phenotype [35]. In contrast, *RPA49* deletion led to a two-fold increase in the accumulation of UPS compared with the pre-rRNA initiated at position +1 (Figure 3A).

To characterize UPS transcripts in wild-type cells in comparison to the *uaf30*∆ mutant, we used a protection assay, which permits the exploration of the chemical status of the 5′ ends of RNAs (Figure 3B). We aimed at differentiating the uncapped 5′-triphosphate (blue) produced by RNAPI from the 5′-end capped RNA (red) resulting from the co-transcriptional modification of the 5′-end of the nascent RNAs produced by RNAPII. In the protection assay, the total RNAs were treated in vitro with polyphosphatase that removed the γ- and β-phosphates from the 5’-triphosphorylated RNAs, and they were subsequently treated with 5′-3′ exoribonuclease, which degraded the 5′-monophosphorylated RNAs. The RNAs were quantified with RT-qPCR, and the ratios of the treated/untreated samples were calculated. Capped RNAs are insensitive to polyphosphatase and exoribonuclease treatments, as shown after using *PMA1* mRNA (Figure 3C, left panel). In contrast, RNAPI transcripts such as +1 TSS rRNA with the triphosphate at their 5′-end were efficiently removed (Figure 3C, middle panel). However, in the *uaf30*∆ strain, a slightly higher fraction of +1 TSS resisted the treatment compared to the wild type, which is fully consistent with the drastic increase of the 5′-extended rRNAs in this mutant (about 20%). The UPS detected in the WT and *uaf30*∆ are both stable in the protection assay (Figure 3C, right panel), with 90% of UPS resisting the treatment, as was the case for the *PMA1* mRNA.

In order to address the polyadenylation status of UPS, we performed reverse transcription using polyT primer and quantified the results with qPCR. We used total RNAs and, to focus on the UPS rRNA produced by RNAPII, we also used RNAs treated with polyphosphatase and exoribonuclease. Only 5% of +1 TSS pre-rRNA transcripts amplified with polyT were resistant to the protection assay in the wild-type cells versus almost 40% in the *uaf30*∆ strain (Figure 3D). We observed the protection of almost 90% of the polyadenylated UPS in WT and 70% in *uaf30*∆ (Figure 3D). Altogether, our experiments indicated that the majority of UPS are RNAPII-dependent transcripts, which are capped and polyadenylated.

### 2.4. UPS Accumulate in Various Growth Conditions

rRNA synthesis and processing are tightly regulated, and they respond to changes in growth conditions. Different stress situations (e.g., temperature shifts, drugs) or growth-limiting conditions (such as stationary phase, post-diauxic shift, starvation) downregulate rDNA transcription in order to adapt cellular resources [36]. During stress response and adaptation, cells change their transcriptional program through chromatin remodeling and the activation of certain transcriptional factors [37]. To regulate rDNA transcription, mammalian cells specifically express lncRNAs [38].

To address UPS transcriptional regulation, we quantified the UPS levels in various growth conditions with RT-qPCR. In agreement with the previous report from the Nomura lab, we observed a 2-fold accumulation of UPS after a shift to 37 °C [28] (Figure 4A). In contrast, the UPS level decreased 2-fold when the cultures were incubated at a lower temperature (24 °C instead the optimal 30 °C). These results indicated that UPS transcription and/or decay are regulated in response to changes in growth conditions. In order to analyze the impact of natural growth-limiting conditions, we used cultures at the post-diauxic phase of growth and at the stationary phase (also called quiescent cells). After a short lag phase (diauxic shift), which appeared once the cells had consumed all of the fermentable carbon sources, the yeast cells switched to the respiration of ethanol and entered into the post-diauxic phase. After 11 days of culture, corresponding to about 180 generations, all of the nutrients had been exhausted, and the cells entered into a state of dormancy [39,40]. Even though the cells in the stationary culture were rather heterogeneous, they were characterized by their repressed transcription and by the presence of a lytic enzyme-resistant cell wall. We extracted RNA from post-diauxic cells (OD_600_ = 2.5–3.0) and 11-day old cultures. The UPS were quantified using RT-qPCR. As shown in Figure 4B, UPS started to accumulate during the post-diauxic phase and remained accumulated in quiescent cells. This result suggests that UPS could participate in the downregulation of rDNA transcription in response to environmental stimuli.

### 2.5. UPS Are Produced from Closed rDNA Repeats

To address the link between UPS transcription and the state of chromatin at the rDNA locus, we used strains with the copy number of rRNA genes stabilized by the deletion of the *FOB1* gene [41]. The used strains contain a stable number of 190 or 25 rDNA copies. The copy number has important implications for RNAPI-dependent transcription and the chromatin state [28,42,43]. In a 25-copy strain, all copies are heavily charged with RNAPI and open, while the 190-copy (WT rDNA copy number) strain contains a similar amount of open and closed copies [9,44]. We measured the amount of UPS and 1+ TSS pre-rRNA in those strains using RT-qPCR. The level of UPS decreases more that 15-fold in the strain with only 25 copies of rRNA gene compared to the 190-copy strain (Figure 5A). This decrease is not a direct consequence of the lower copy number since +1 TSS pre-rRNA remained unchanged. We suggest that in conditions of high RNAPI occupancy, the RNAPII has little or no access to the promoter region. The direct consequence of this assumption is that the RNAPII may play a role in establishing or maintaining the closed state.

To address this question, we performed time-course experiments in cultures entering and exiting the stationary growth phase (Figure 5B). We analyzed cells obtained from the exponentially growing culture (EXPO), 24 h culture, 48 h culture (post-diauxic), and STAT (11 days; stationary phase sample). Cells from STAT were diluted with fresh media containing glucose, and samples were taken every 60 min up to 6 h (Figure 5C, red). Despite the high experimental variation between samples, we could show that UPS progressively accumulated during cell growth in agreement with our data at the post-diauxic phase and the stationary phase (compare Figure 4B and Figure 5C). The amount of UPS increased in quiescent cells, while +1 TSS accumulation decreased by about 20 fold (Figure 5C, compare red and blue). Subsequently, after adding sugar into the stationary phase culture, the UPS quantity decreased progressively to the low levels that had been identified in the exponential growing cells (Figure 5D). At the same time, pre-rRNA produced from +1 TSS (*i.e.*, produced by RNAPI) progressively decreased during growth to the stationary phase and regularly increased after the return to exponential growth (Figure 5D). This result suggests that UPS may participate in the regulation of RNAPI transcription, and that it might occur by participating in the process to control the chromatin state.

## 3. Discussion

Genomes are pervasively transcribed, and various RNA are produced [45,46,47]. Due to the unstable nature and low abundancy of some ncRNAs in the cell, their characterization in wild-type cells is challenging, and they have mostly been discovered using mutants or particular growth conditions. For example, several classes of ncRNAs in yeast were identified in RNA decay mutants (*i.e.*, CUTs, XUTs, NUTs; [30,31,48,49,50] or in meiosis for MUTs; [51]). rRNAs represent more than 90% of the cellular RNAs and are specifically removed in the preparation of classical libraries. That is why the produced ncRNAs that potentially form the rDNA locus could not be found using the classical high-throughput sequencing analyses that led to the characterization of lncRNA classes. In addition, the complex organization of the rDNA locus, composed of about 100–150 copies of the rDNA gene, and its tremendous transcription rate hinder its functional dissection [52].

Among ncRNAs, widespread long non-coding RNAs potentially regulate gene expression in *cis* (transcriptional interference, scaffolding at transcriptional site) or in *trans* (at any other cellular location [53]). Although some lncRNAs have clearly established roles, the vast majority remains uncharacterized. Here, we have further characterized the lncRNAs produced from the RNAPII-dependent promoter located upstream from +1 TSS of 35S rDNA (Figure 1B). Although these 5′-extended transcripts are abundant in *uaf30*∆ and PSW mutants [19,20], they have never been studied in wild-type cells. These ncRNAs are not the Trf4-sensitive lncRNAs starting from IGS1 described in [54] nor analyzed by [43] C-pro—initiating antisense transcripts. However, the bidirectionality of C-pro should not be excluded, as described for E-pro in the IGS1, to promote UPS synthesis. Here, we optimized RT-qPCR experiments to detect and quantify UPS. RNAPII starts transcription in the region from 90 to 9 bp upstream from +1TSS to produce UPS. In our experiments, we detected at least 10 different rRNA species (from −70 to −11, Appendix A), suggesting multiple initiation events at potentially each of the 100–150 promoters or specific RNAPII start sites in different copies. Nevertheless, we cannot rule out the 5′-end processing events of these transcripts. UPS accumulated under the culturing of wild-type cells at high temperatures (shown by [28] and confirmed here) and at the post-diauxic and stationary phases. Despite the lack of agreement about the relevance of ncRNAs, especially those presenting at very low levels, some of them have an attributed functionality. MUTs, shown to regulate meiosis in *S. cerevisiae* [51] and a few other lncRNA: *PHO84* [55]; *GAL1-10* upstream ncRNA [56,57], or TERRA [58], could play a role in the adaptation to growth condition changes. We propose UPS as a lncRNA belonging to this last category.

Using the mutants of RNA decay pathways, we have shown that exosomes and their co-factor TRAMP complex do not affect the accumulation of UPS. However, mutations in both Xrn1 and Rat1, 5′-3′ exoribonucleases have increased UPS levels when mutated or absent. This result has interesting implications; according to the torpedo/allosteric model, Rat1 could be associated with the elongation of RNAPII at the 35S gene and could promote UPS termination, as demonstrated for mRNAs [59]. Its presence at the rDNA locus would facilitate immediate UPS degradation. Although Rat1 and Xrn1 contribute to the co-transcriptional degradation of nascent RNAs, this degradation might be insufficient to cause polymerase release [59]. Moreover, Rat1 could function in 3′-end processing by enhancing the recruitment of 3′-end processing factors, including Pcf11 and Rna15 [59]. The participation of these factors in UPS degradation needs to be established. According to their accumulation in the Xrn1-lacking strain, UPS could be classified as a XUT/SUT [31,47]. However, XUTs are mostly antisense long non-coding (aslnc)RNAs, which possess a sense protein-coding mRNA counterpart [60]. The existence of a UPS fraction accumulating in *xrn1*∆, though degraded by Xrn1, suggests that these transcripts escape Rat1-dependent degradation after transcription termination. Subsequent degradation by Xrn1 could provide a double safety mechanism. Altogether, this suggests that the transcription event is important by itself, but is also suggests that UPS could act in *trans*.

Pervasive transcription as well as its output—the ncRNA, could become important in particular conditions. As mentioned, UPS accumulated in cells cultured at high temperatures, in post-diauxic shift, and in the stationary phase. In these cells, transcriptional programs change radically. Furthermore, in the stationary phase, the cell growth is arrested and rRNA synthesis decreases, shifting from a productive to non-productive rRNA processing pathway [61]. The accumulation of UPS in these conditions indicates that the repressed state of RNAPI transcription and rDNA chromatin could be prone to RNAPII transcription and UPS production. Previous studies have addressed the repressive function of RNAPI on RNAPII transcription, which could represent the natural control of ncRNA production. For example, H3 and H4 histone acetylation at the ncRNA cryptic promoters (E-pro and C-pro; [43]) would change chromatin state and its accessibility for RNAPII machinery. In agreement with these results, we have also found that increased UPS levels correlate to the closing of rDNA genes [9]. We have observed that the UPS level was very low in a strain bearing only 25 rDNA copies heavily occupied by RNAPI [42]. As previously proposed, heavy loading by RNAPI can drive the silencing of RNAPII transcription [28]. Since half of the rDNA genes in the wild-type strains are found in the closed chromatin state, it is strongly suggested that in WT strains, UPS are produced from closed copies. How copies are changing their chromatin state and accessibility depends on the RNAPI and Hmo1 [44]. However, we speculate that RNAPII producing UPS could participate in the regulation of the chromatin state. In our time course experiments, in which the chromatin state changed from open to closed rDNA copies, UPS accumulated in cells at the stationary phase. Once the cells were released from dormancy by the addition of sugar, the UPS level decreased and correlated to the progressive increase in rRNA production and the recovery of cell growth. Although these results suggest that UPS are mainly produced from closed copies by RNAPII, we cannot rule out a first wave of RNAPII transcription participating in the closing of chromatin at the rDNA locus. At least two scenarios could be proposed here:**(1)** UPS produced at a low level from closed rDNA copies by RNAPII-dependent transcription are important for the maintenance of the closed chromatin state. In response to environmental stimuli, the recruitment of the UAF complex represses the RNAPII that disengage from the rRNA promoter;**(2)** The RNAPII transcribing rRNA, revealed in wild-type cells by the presence of UPS, provides a platform for the elongation factors regulating rRNA production. Most of them are not essential for basal RNAPI activity but could be crucial for nucleosome eviction/assembly, chromatin remodeling during stress, and adaptation. Such recruitment could regulate the replication-independent closing of open rDNA genes [9,62].

To date, we are not aware of any easy methods to directly study the effects of ncRNA *i.e.*, ncRNA deletion or over-expression, because their genes mostly overlap with other protein-coding genes. In the case of the rRNA, the situation is even more complicated because of the coexistence of multiple copies in different chromatin states. We have shown here that in wild-type cells in permissive/optimal growth conditions, in addition to RNAPI, RNAPII transcribes the rDNA to produce lncRNAs—specially, UPS. The role of these transcripts as well as the fine characterization of their promoter awaits further investigation.

## 4. Materials and Methods

### 4.1. Yeast Strains and Plasmids

The strains used in this study are described in Appendix A. The experiments were mostly performed with the derivatives of BY4741 (S288C, Euroscarf, Frankfurt, Germany) or W303 [63] backgrounds. Double mutants were constructed by crossing single mutant strains after sporulation tetrads were analyzed and double mutant selected. Gene deletion was verified with PCR using upstream and downstream external primers.

### 4.2. Media and Culture Conditions

Growth media and plates were prepared with standard methods using YPD containing 2% glucose. Standard growth conditions were employed (30 °C, shaking 180 rpm), and heat shock was performed at 37 °C for an indicated time. For galactose growth, YPGal medium (2% galactose) was used. The time course experiment was performed in YPD media at 30 °C. 500 mL culture was grown, and samples for RNA extractions were collected at indicated time-points. After 11 days of culturing, cells were considered as stationary. After centrifugation, all cells were resuspended in fresh YPD medium, and each 60 min, samples were collected for RNA analysis. For the *pMET3-RAT1* repression experiment, cells were pre-cultured ON in YPD at 30 °C. ON exponential cultures were used as the 0 time-point, and after careful centrifugation, cells were shifted to YPD medium containing 6 mM methionine. After 12 h of incubation, cells were centrifugated and collected for RNA analysis.

### 4.3. RNA Extraction

Total RNA was extracted using the hot phenol extraction method from 10- to 50-mL of cultures at OD_600_ = 0.4–0.8, followed by DNA contaminant removal using RNase-free DNaseI. RNA concentration was measured on a nanodrop spectrophotometer. For the protection assay and *pMET3-RAT1* experiment, RNA extractions were performed as previously described [64].

### 4.4. Primer Extension

The RT reaction was performed using the same method as the standard reverse transcription reaction but with radiolabeled RT primer (Figure 1A). 100 ng and 1 μg of total RNA was used for PSW, *uaf30*∆, and wild type. The total volume of the RT reaction was loaded on a denaturing 10% acrylamide gel. The gel was subsequently dried and exposed.

### 4.5. Reverse Transcription and qPCR (RT-qPCR)

For the RT reaction, 1 μg of total RNA was subjected to reverse transcription (Go Reverse Transcriptase, Promega, Madison, WI, USA) with a mix of locus-specific (target and control genes) or random primers for 60 min at 42 °C. cDNA were quantified with real-time qPCR using iTaq universal SYBR Green Supermix (Bio-Rad, Hercules, CA, USA) and the ViiA7 AB Applied Biosystems (Life Technologies, Carlsbad, CA, USA). Primer pairs used for amplification are listed in Appendix A. Signals were analyzed with Quant Studio Real Time PCR Software v1.1 and were normalized with 35S rRNA (TSS) and *scR1* signals. Error bars correspond to standard deviation over at least two or more independent cultures.

## Figures and Tables

**Figure 1 ncrna-07-00041-f001:**
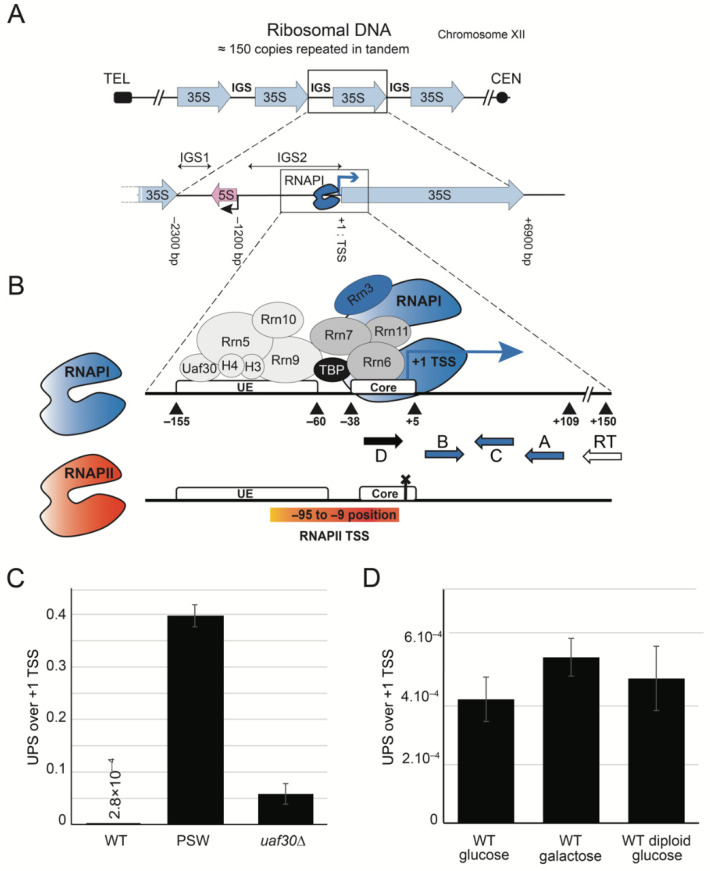
UPS lncRNAs are transcribed from the IGS2 region in rDNA. (**A**) Schematic representation of the rDNA locus in yeast *Saccharomyces cerevisiae*. The budding yeast contains a single rDNA cluster organized as a linear array of 100–150 rDNA repeats on chromosome XII. Each repeat, of 9.1-kb, produces a 35S precursor rRNA (35S gene is the blue arrow, sequence coordinates are relative to +1 TSS). IGS (intergenic sequence) is separated by the 5S gene (transcribed by RNAPIII, red arrow) into IGS1 and IGS2. (**B**) Details of the 35S promoter region composed of UE (Upstream Element) and core promoter. UE is occupied by the UAF complex (Uaf30, Rrn5, Rrn9, Rrn10, H3, and H4 subunits represented as light grey ovals). Core promoter is occupied by the Core Factor (CF) complex (Rrn6, Rrn7, and Rrn11 represented as grey ovals). UAF and CF are connected by TBP (black ovals). RNAPI bound to Rrn3 (in blue) binds CF bound to the promoter to form the pre-initiation complex. Primers used in experiments are represented below; RT forward primer was used for primer extension and reverse transcription, A-B and C-D couples were used in qPCR. Putative RNAPII (in red) promoter region (−95/−5) overlapping UE and core is represented as an orange rectangle. (**C**) UPS accumulated in PSW and *uaf30*∆ strains and are present at a very low level in wild-type strain. 1 μg of total RNA was used for reverse transcription using RT primer. cDNAs were quantified by qPCR using the C-D couple. Detected quantities of 5′-extended transcripts were normalized by the values of A-B amplifications corresponding to +1 TSS (+1/109 bp). Error bars correspond to standard deviation of three independent cultures. (**D**) Quantification of UPS in wild-type strains. Wild-type cells (BY4741) were cultivated overnight to the exponential phase in rich media containing glucose or galactose as a carbon source. Diploid strain (BY4743) was cultivated in glucose at 30 °C. UPS were quantified using RT primer for reverse transcription (1 μg of total RNA), C-D couple for UPS, and A-B couple for +1 TSS. Error bars correspond to the standard deviation of two independent cultures.

**Figure 2 ncrna-07-00041-f002:**
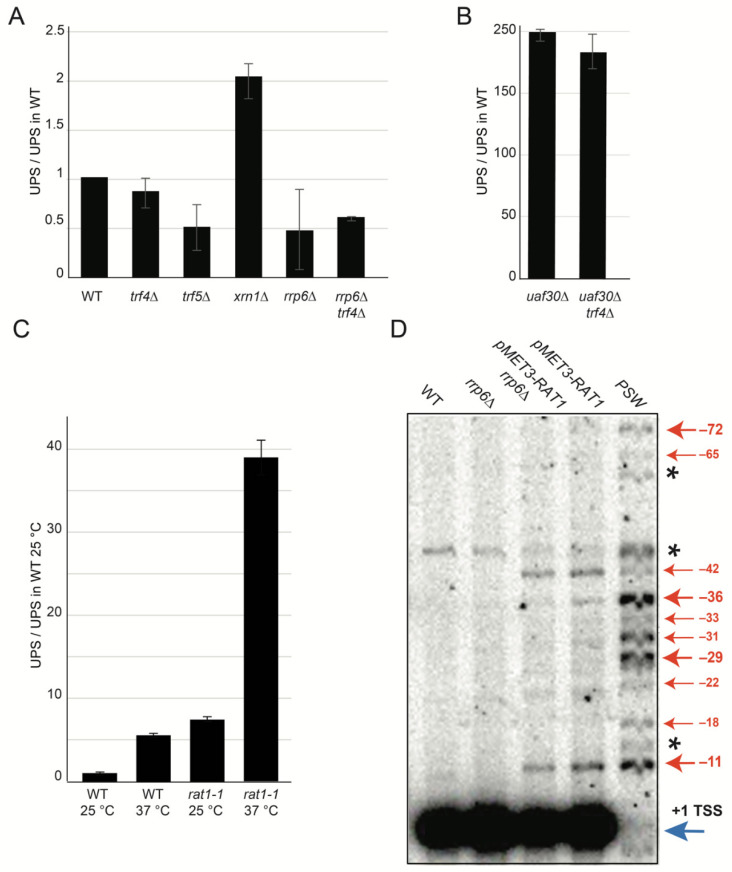
UPS are unstable lncRNAs degraded by Xrn1 and Rat1 5′-3′ exoribonucleases. (**A**) UPS accumulate in *xrn1*∆ but not in exosomes (*rrp6*∆) or in the TRAMP complex (*trf4*∆, *trf5*∆) mutants. Yeast cells were exponentially grown in YPD medium at 30 °C overnight. Quantification of UPS in RNA decay mutants using qPCR and primers are described in Figure 1B. Quantities of UPS in mutants are represented as a ratio to UPS quantities in wild type. Error bars correspond to the standard deviation of two or three independent cultures. (**B**) UPS accumulation in *uaf30*∆ and *uaf30*∆ *trf4*∆ mutants. UPS were quantified as described in (**A**). (**C**) UPS accumulate in *rat1-1* mutated strain. WT and *rat1-1* strains (BY4741) were grown overnight at 25 °C to early exponential phase and then shifted to 37 °C for 6 h. Quantification of UPS in RNA decay mutants using qPCR. Quantities of UPS in mutants are represented as a ratio to UPS quantities in wild type at 25 °C. Error bars correspond to standard deviation of two independent cultures. (**D**) Visualization of UPS in Rat1-depleted cells. Cells were cultivated in YPD medium than switched to YPD+ 6 mM methionine for 12 h. Primer extension reaction with radiolabeled RT primer (Figure 1B) was resolved in 10% denaturating acrylamide gel. UPS are marked by red arrows and +1 TSS is marked by the blue one (please note the saturated exposure of +1 TSS signal). Star (*) represents non-specific RT product.

**Figure 3 ncrna-07-00041-f003:**
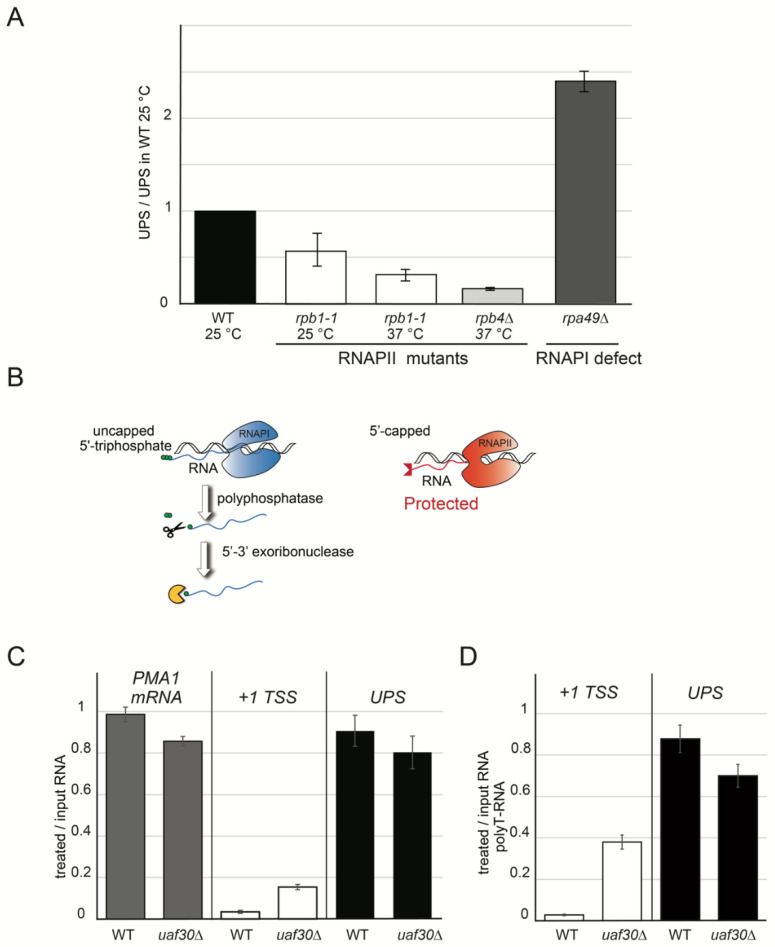
UPS are produced by RNAPII. (**A**) Quantities of UPS decreased in RNAPII mutants. RT-qPCR was performed on total RNA extracted from exponentially grown cultures grown at 25 °C overnight shifted to 37 °C for 1 h for *rpb1-1* mutant. *rpb4*∆ and *rpa49*∆ mutants were processed at 30 °C. *rpb1-1* and *rpb4*∆ mutations provoke defects on RNAPII, and *rpa49*∆ was used for RNAPI repression conditions. Reverse transcription and qPCR were performed as in Figure 1C for UPS, but specific scR1-R primer was added for internal normalization. Error bars correspond to the standard deviation of three independent cultures. All values were normalized by WT 25 °C values. (**B**) Principles of RNA protection assay. Transcribing RNAPII is in red, and RNAPI is in blue. Terminal 5′-triphosphates of RNAPI-specific transcripts are presented as green circles and RNAPII-specific cap as red triangles. Subsequent steps of in vitro treatment are as follows: polyphosphatase depicted as scissors and 5′-3′ exoribonuclease as a yellow Pacman. (**C**) UPS transcripts are capped. Total RNA from wild-type and *uaf30*∆ mutated strains (BY4741) were treated as depicted in (**B**). Reverse transcription was performed using random primers. UPS and +1 TSS were quantified as described in Figure 1C, and *PMA1* mRNA was amplified with its specific primers. Values are depicted as ratios of treated versus untreated RNAs. Error bars represent the standard deviation of three independent measurements. (**D**) UPS transcripts are polyadenylated. The experiment and quantification were performed as in (**C**) but reverse-transcribed using polyT oligonucleotides in order to amplify polyadenylated transcripts.

**Figure 4 ncrna-07-00041-f004:**
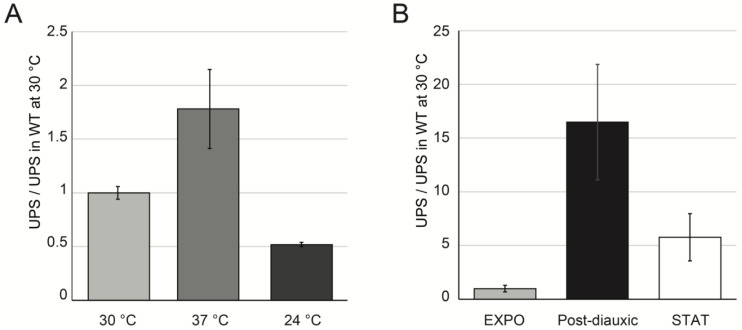
UPS transcripts accumulated in various growth conditions. (**A**) High temperature increased UPS level. Total RNA was extracted from wild-type cells cultivated overnight to exponential phase in YPD at indicated temperatures. UPS level was quantified as in Figure 1C. Error bars correspond to the standard deviation of three independent cultures. (**B**) UPS accumulated in post-diauxic and stationary (STAT) wild-type cells. Cells were grown in YPD at 30 °C overnight to the exponential phase (EXPO) and then left for either an additional 24 h or 11 days. RNA extraction and experimentation was performed as in Figure 1C, data were normalized by *scR1* RNA reverse transcribed with specific primer (scR1-R),and quantified with an scR1F/R couple. *scR1*-normalized values were represented as a ratio of WT exponential values. Error bars correspond to the standard deviation of three independent cultures.

**Figure 5 ncrna-07-00041-f005:**
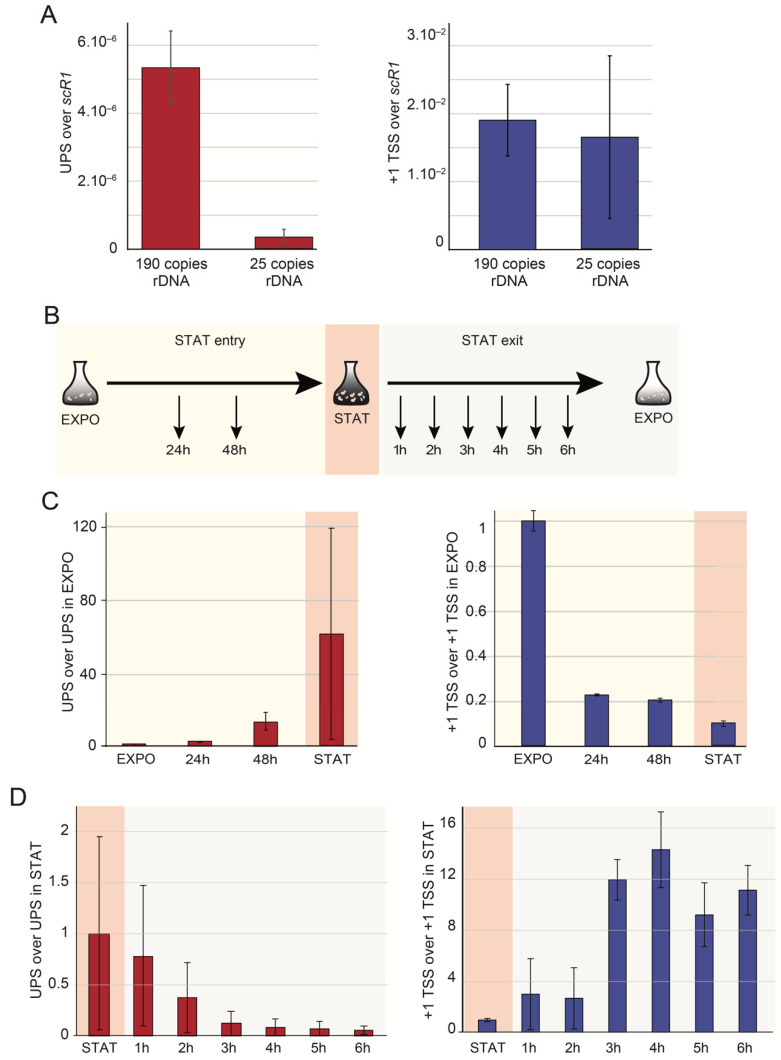
UPS are produced from closed rDNA copies, accumulated in quiescent cells, and anti-correlated with +1 TSS rRNA. (**A**) UPS levels were higher in 190-copy rDNA strains when compared to those with 25 rDNA copies. Strains bearing 190 or 25 rDNA copies were grown to the exponential phase (ON) at 30 °C in YPD media. RNA was extracted, reverse transcribed with RT and sCR1-R primers, and subsequently quantified as in Figure 1C. Both UPS (red) and +1 TSS (blue) are shown. (**B**) Schematic representation of time-course experiment. Cells were grown overnight to the exponential phase, samples were collected (EXPO) then incubated over the following 24 and 48 h when the post-diauxic samples were collected (STAT entry). Afterwards, cultures were cultivated for additional 9 days (11 days of culture in total, STAT) and collected. Stationary cells were then resuspended in fresh YPD medium and cultivated for 6 h (STAT exit). Each hour, cells were collected for RNA analysis. (**C**) UPS accumulate in the post-diauxic stage and stationary culture. Total RNA was extracted and quantified as in Figure 1C. For easier visualization, values for 24 h, 48 h, and STAT samples were normalized to UPS and +1 TSS measured in the exponentially growing culture. Both UPS (red) and +1 TSS (blue) are shown. (**D**) UPS and +1 TSS accumulation are anti-correlated when cells resume exponential growth. Stationary cells were resuspended in fresh medium. Error bars correspond to the standard deviation of two independent cultures.

## Data Availability

All data in this study are included in this published article and its Appendix A.

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
