# Peer review of "Non-Coding, RNAPII-Dependent Transcription at the Promoters of rRNA Genes Regulates Their Chromatin State in S. cerevisiae"

_ncrna, 2021, doi:10.3390/ncrna7030041_

Round 1

Reviewer 1 Report

Lesage et al describe the detection of 'upstream initiating transcripts' (UPS) from the rDNA locus which presumably originate from 'closed' rDNA copies. They show that these transcripts seem to be transcribed by RNA Pol II, capped and poly-adenylate. Furthermore, they show that these transcripts seem to be upregulated in stationary phase. Overall, the experiments are well described and support the conclusions. 

Comments to the authors: 

line 134: I think this refers to S1C, not S1D ? (I could not find S1D)
line 136-7: please show these control experiments

Figure 3D: could the authors please explain why the RNAs are treated with pyro-phosphatase / exonuclease for this experiment ? Would it not be sufficient (and therefore cleaner) to compare the pA / total for the different RT-qPCR primer pairs ?

line 273 ff: the occurrence of the UPS in post-diauxic and quiescent cells is highly interesting. It would strengthen the manuscript if the authors would verify that also these UPS are capped / poly-adenlyated; verification that they are indeed Pol II dependent is probably difficult ?

Figure 5 A, C, D: the layout of mixing blue and red with different y-axis in one plot is confusing. I would suggest to split the graphs, also to make the design consistent with the other figures. 

Figure 5A: it would be great to add 'wild-type' cells (as used in the other experiments) for comparison

Author Response

Reviewer 1

Lesage et al describe the detection of 'upstream initiating transcripts' (UPS) from the rDNA locus which presumably originate from 'closed' rDNA copies. They show that these transcripts seem to be transcribed by RNA Pol II, capped and poly-adenylate. Furthermore, they show that these transcripts seem to be upregulated in stationary phase. Overall, the experiments are well described and support the conclusions. 

We would like to thanks reviewer for his comments. Please find our detailed answer bellow:

line 134: I think this refers to S1C, not S1D ? (I could not find S1D)

This have been modified to S1C.

line 136-7: please show these control experiments

We have now included a panel S1D including those experimental results.

Figure 3D: could the authors please explain why the RNAs are treated with pyro-phosphatase / exonuclease for this experiment ? Would it not be sufficient (and therefore cleaner) to compare the pA / total for the different RT-qPCR primer pairs ?

It is well established that RNAPI transcribed rRNA can be poly-adenylated: rRNA are targeted to degradation by TRAMP complex, which contains two polyA-polymerase (Trf4/Trf5; see Dez et al., Embo, 2006). To establish poly-adenylation status of UPS rRNA, this treatment is mandatory. We have now clarified this point in the text (see line 244/246).

line 273 ff: the occurrence of the UPS in post-diauxic and quiescent cells is highly interesting. It would strengthen the manuscript if the authors would verify that also these UPS are capped / poly-adenlyated; verification that they are indeed Pol II dependent is probably difficult ?

We agree that this experiment would have been of interest. To accurately evaluate this protection experiments, we use control mRNA degradation efficiency. Unfortunately, massive variation of transcriptome in quiescent cells, making mRNA control experiment difficult to interpret and not reproducible. As mentioned by the reviewer, this is technically challenging, and we could not explore UPS in protection assay.

Figure 5 A, C, D: the layout of mixing blue and red with different y-axis in one plot is confusing. I would suggest to split the graphs, also to make the design consistent with the other figures. 

We have followed reviewer suggestion, and now present data in different panel

Figure 5A: it would be great to add 'wild-type' cells (as used in the other experiments) for comparison

Wild-type is comparable to 190 copies strains (both for UPS and +1 TSS), but experiments were not performed in fully comparable condition.

Reviewer 2 Report

In the manuscript entitled “Non-Coding, RNAPII-Dependent Transcription at the Promoters of rRNA Genes Regulates Their Chromatin State in S. cerevisiae”, the authors characterized long non-coding RNAs generated from ribosomal gene promoter, called the UPStream-initiating transcripts (UPS). These are present in exponentially growing wildtype cells in very low levels and degraded by the Xrn1 and Rat1 exonuclease. These Pol II transcripts are capped and polyadenylated and likely produced from the closed copies of rDNA genes. In general, the authors provide strong evidence for their conclusions and characterize this ncRNA in molecular detail. As the authors also discuss, it will be important to evaluate the functional importance of this RNA species and I hope my suggestions below can help a bit. I have a few other suggestions and comments that need to be addressed before publication. Major comments • In Figure 1, the authors show the general detection strategy for UPS using RT-qPCR with specific primers that only cover Pol I transcribed 35S rRNA as a control (primer pair A-B) versus extended PCR fragments into the putative Pol II promoter region (primers A-C, A-D and A-E). Have the authors tested the different primer efficiencies in the different combinations? My concern is that the different levels of 5’ extended rRNAs are affected by these different amplicon sizes in the PCR reaction, which makes the quantification difficult in my opinion. • Related to this, I do not understand the conclusion from Figure 1C that “In uaf30∆ and PSW strains, 5’ extended 35S pre-rRNAs represent 10% and 45% of 35S pre-RNA, respectively”. Line 130. The three amplicons to detect UPS are partially overlapping so it seems that summing up the fractions results in redundant counting of the transcripts, for example amplicon A-C is also part of the amplicons A-D and A-E, right? • The authors mention in the text that they have made several important controls to make sure the detection of UPS is far from background detection (reaction without reverse transcriptase or negative control such as mRNAs targeted against deleted genes). These are important controls and I would suggest to add them into the manuscript as an additional supplementary Figure. • Figure 3 shows by elegant genetic biochemical experiments that majority of UPS are RNAPII-dependent transcripts, which are capped and polyadenylated in contrast to Pol I transcripts. These data are convincing and support the main conclusions. • Figure 4 shows differences in UPS levels at different growth phases and temperatures and indicate changes in the UPS levels at higher temperature and post-diauxic and stationary phase. Can the authors correlate the UPS levels with the ratio of open versus closed rDNA copies (by e.g psoralen assays) to check whether this increase can be also explained by a higher fraction of closed copies? • Figure 5 shows that UPS levels are very low in a low-copy rDNA strain, suggesting that these transcripts are produced from the closed repeats. Timecourse experiments suggest that UPS levels and +1 TSS RNA levels anticorrelate upon growth in stationary phase and release. The authors suggest that UPS levels are therefore derived from the closed repeats. This argument is a bit indirect as the authors also discuss later that it cannot be excluded that these UPS levels are generated for example during the conversion of a repeat from a closed to an open state. To get more functional insights into the role of UPS in regulating rDNA chromatin state, the authors could repeat the growth arrest and release experiment in the rat1-1 or rpb4 delta backgrounds where the authors show a massive up or down regulation of UPS. It will be interesting to see what effect this has on the opening and closing of rDNA repeats (for example by psoralen assays). Minor comments: • Figure 2 shows convincingly that UPS are unstable lncRNAs degraded by Xrn1 and mostly Rat1 5’-3’ exoribonucleases. The primer extension analysis shows clearly 5’ extended transcripts at -11 and -42 position in the rat1 mutant background. Can the authors remove the false-color in the blot? I understand the authors wanted to show the overexposure situation for the +1 TSS, but I don’t think is necessary/crucial. • Line 83. Check this sentence: “If the function of those transcripts remains elusive, they were instrumental to explore RNAPI activity.”

Author Response

Reviewer 2

                         In the manuscript entitled “Non-Coding, RNAPII-Dependent Transcription at the Promoters of rRNA Genes Regulates Their Chromatin State in S. cerevisiae”, the authors characterized long non-coding RNAs generated from ribosomal gene promoter, called the UPStream-initiating transcripts (UPS). These are present in exponentially growing wildtype cells in very low levels and degraded by the Xrn1 and Rat1 exonuclease. These Pol II transcripts are capped and polyadenylated and likely produced from the closed copies of rDNA genes. In general, the authors provide strong evidence for their conclusions and characterize this ncRNA in molecular detail. As the authors also discuss, it will be important to evaluate the functional importance of this RNA species and I hope my suggestions below can help a bit. 

I have a few other suggestions and comments that need to be addressed before publication. 

Major comments •

 In Figure 1, the authors show the general detection strategy for UPS using RT-qPCR with specific primers that only cover Pol I transcribed 35S rRNA as a control (primer pair A-B) versus extended PCR fragments into the putative Pol II promoter region (primers A-C, A-D and A-E). Have the authors tested the different primer efficiencies in the different combinations? My concern is that the different levels of 5’ extended rRNAs are affected by these different amplicon sizes in the PCR reaction, which makes the quantification difficult in my opinion. Related to this, I do not understand the conclusion from Figure 1C that “In uaf30∆ and PSW strains, 5’ extended 35S pre-rRNAs represent 10% and 45% of 35S pre-RNA, respectively”. Line 130. The three amplicons to detect UPS are partially overlapping so it seems that summing up the fractions results in redundant counting of the transcripts, for example amplicon A-C is also part of the amplicons A-D and A-E, right?

As aptly pointed by the referee comparison of different amplicons is a complicated issue. Indeed, A-C, A-D and A-E amplicons (mentioned in previous version of the manuscript) overlap and are of different sizes. It should also be noted that rRNA is a difficult template due to its secondary sequences and that IGS DNA sequence contains repetitions and motifs making challenging its efficient quantification.

In qPCR, we use external standard curve (ten-fold dilutions of sonicated genomic DNA for WT strain).

After careful verification of our data, we have noticed that efficiency of -50 amplicon was lower and that for -75 amplicon two couples tested (shown below) give the same efficiency. In order to clarify our conclusion, we show only -26 values.

>YNCL0020C_S288C RDN37-2 SGD:S000006487

rRNA gene 35S ribosomal RNA (35S rRNA) transcript

AGAGGGCAAAAGAAAATAAAAGTAAGATTTTAGTTTGTAATGGGAGGGGGGGTTTAGTCA

TGGAGTACAAGTGTGAGGAAAAGTAGTTGGGAGGTACTTCATGCGAAAGCAGTTGAAGAC

AAGTTCGAAAAGAGTTTGGAAACGAATTCGAGTAGGCTTGTCGTTCGTTATGTTTTTGTA

AATGGCCTCGTCAAACGGTGGAGAGAGTCGCTAGGTGATCGTCAGATCTGCCTAGTCTCT.   R1

ATACAGCGTGTTTAATTGACATGGGTTGATGCGTATTGAGAGATACAATTTGGGAAGAAA

TTCCCAGAGTGTGTTTCTTTTGCGTTTAACCTGAACAGTCTCATCGTGGGCATCTTGCGA

TTCCATTGGTGAGCAGCGAAGGATTTGGTGGATTACTAGCTAATAGCAATCTATTTCAAA

Two yellow primers amplify - 75 couple 1

Yellow primer F and green R1 amplifie -75 couple 2

Two green primers amplify -50 product

  • The authors mention in the text that they have made several important controls to make sure the detection of UPS is far from background detection (reaction without reverse transcriptase or negative control such as mRNAs targeted against deleted genes). These are important controls and I would suggest to add them into the manuscript as an additional supplementary

We have now included reaction without reverse transcriptase in figure S1D.

  • Figure 3 shows by elegant genetic biochemical experiments that majority of UPS are RNAPII-dependent transcripts, which are capped and polyadenylated in contrast to Pol I transcripts. These data are convincing and support the main conclusions. 

  • Figure 4 shows differences in UPS levels at different growth phases and temperatures and indicate changes in the UPS levels at higher temperature and post-diauxic and stationary phase. Can the authors correlate the UPS levels with the ratio of open versus closed rDNA copies (by e.g psoralen assays) to check whether this increase can be also explained by a higher fraction of closed copies?

Careful exploration of open versus closed rDNA have been performed previously (see Wittner et al., Cell, 2011). We could observed that open/closed fraction depends on genetic background as shown below for BY 4741 strain. After 11 days of culture (stationary phase) the ratio of open to closed chromatin is rather 40/60 when compared to exponentially grown cells (50/50).

  • Figure 5 shows that UPS levels are very low in a low-copy rDNA strain, suggesting that these transcripts are produced from the closed repeats. Timecourse experiments suggest that UPS levels and +1 TSS RNA levels anticorrelate upon growth in stationary phase and release. The authors suggest that UPS levels are therefore derived from the closed repeats. This argument is a bit indirect as the authors also discuss later that it cannot be excluded that these UPS levels are generated for example during the conversion of a repeat from a closed to an open state. To get more functional insights into the role of UPS in regulating rDNA chromatin state, the authors could repeat the growth arrest and release experiment in the rat1-1 or rpb4 delta backgrounds where the authors show a massive up or down regulation of UPS. It will be interesting to see what effect this has on the opening and closing of rDNA repeats (for example by psoralen assays). 

We agree with reviewers comments. However, growth arrest following starvation, and stationary exit is difficult to reproduce in wild-type cells (see high error bar in experiments presented in Figure 5C and Figure 5D). rpb4∆ and rpb1-1 mutants present clear growth defect in exponentially growing cells, making such experiment very difficult to control. We have ,not perform such experiment.

Minor comments: • Figure 2 shows convincingly that UPS are unstable lncRNAs degraded by Xrn1 and mostly Rat1 5’-3’ exoribonucleases. The primer extension analysis shows clearly 5’ extended transcripts at -11 and -42 position in the rat1 mutant background. Can the authors remove the false-color in the blot? I understand the authors wanted to show the overexposure situation for the +1 TSS, but I don’t think is necessary/crucial. •

We have removed false-color, and saturation is now mentioned in figure legend.

 Line 83. Check this sentence: “If the function of those transcripts remains elusive, they were instrumental to explore RNAPI activity.” 

We modified this sentence.

Round 2

Reviewer 1 Report

Dear authors, thank you for your replies and clarifications. Congratulations on a beautiful paper!

Author Response

We want to thanks reviewers for their very positive and constructive comments, that have improved our manuscript.